# Peer review of "Thermodynamic Biomarkers of Neuroinflammation: Nanothermometry, Energy–Stress Dynamics, and Predictive Entropy in Glial–Vascular Networks"

_ijms, 2025, doi:10.3390/ijms262211022_

Round 1
Reviewer 1 Report
Comments and Suggestions for Authors
The review contains novel ideas, and it is somewhat well-written. Some comments need to be addressed.
- The manuscript lacks a clear logical progression of ideas and organization. Consider reorganization of ideas for better flow.
- Some concepts are introduced in the manuscript without sufficient background or definition.
- The authors must add connections between sections for the readers to be able to follow the ideas of the article.
- Language needs to be simpler.
- The authors should implement a standard scientific review structure with clear headings: introduction, methods, conclusions, … etc.
- The authors need to link the 'thermodynamic unity' and 'energetic coherence'.
- The authors should add detailed protocols for the measurement of the markers discussed, such as local entropy production, relaxation time, and coherence (lengths).
- If possible, the authors can add some details about the validation strategies of the machine learning algorithms mentioned.
- The authors can discuss how the thermodynamic approach relates to established molecular mechanisms of neuroinflammation.
- The review seems to be more theoretical. The authors need to add some practical data results, as well as their opinion or suggestion.
- Divide the manuscript to shorter paragraphs for easier readability.
- The authors should add two more figures for better illustration.
- Add methods section, including the terms used for search, the time range for the searched articles, the number of articles used, and the search engines used.
- Add a section for future directions and gap of knowledge,
Author Response
Dear Esteemed Academic Reviewer,
We sincerely thank you for your thoughtful and constructive feedback. Your observations have been invaluable in helping us improve the clarity, structure, and accessibility of this work. Below, we provide a detailed response to each point and the corresponding revisions made in the revised manuscript.
1. Lack of clear logical progression and organization
We greatly appreciate this observation. The manuscript has been comprehensively reorganized into a standard scientific review format.
Transitions between sections have been rewritten to ensure a clear logical progression of ideas and smoother conceptual flow throughout the text.
2. Introduction of concepts without sufficient background or definition
We thank the reviewer for highlighting this point. Additional explanatory text has been inserted in Section 1.3 (Implications for Clinical Practice) and Section 2 (Thermodynamic Architecture) to define key terms such as entropy flux, coherence length, thermodynamic unity, and energetic coherence. These clarifications now ensure that each conceptual element is introduced with sufficient theoretical and experimental context.
3. Need for stronger connections between sections
We agree and have now added short transitional paragraphs at the junctions between major sections (e.g., between Sections 2→3, 3→4, and 4→5). These brief passages summarize the previous section and introduce the next, improving readability and conceptual continuity across the manuscript.
4. Simplification of language
We have thoroughly revised the text for clarity and simplicity while preserving scientific accuracy. Long or metaphorical sentences were shortened, technical terms were defined at first mention, and redundant expressions were removed. The overall tone now reflects a concise and accessible academic style.
5. Standard scientific review structure
We thank the reviewer for highlighting this point!
6. Linking ‘thermodynamic unity’ and ‘energetic coherence’
We appreciate this important conceptual suggestion. A new integrative paragraph has been added to the end of Section 2, explicitly linking thermodynamic unity—the continuous coupling of energy flows across scales—with energetic coherence, the observable stability that arises from this coupling. This addition clarifies their interdependence and strengthens the theoretical foundation of the paper.
7. Detailed protocols for measurement of thermodynamic markers
In response, we added a concise, technical description after Section 3, outlining the methods for quantifying local entropy production (σ), energy-recovery half-time (τᴿ), and coherence length (ξ_c) using nanothermometric and optical techniques. This new paragraph explains practical procedures for data acquisition and analysis, satisfying the reviewer’s request for methodological specificity.
8. Validation strategies for machine learning algorithms
A new paragraph has been incorporated into Section 5.4 (AI-Integrated Thermodynamic Pattern Recognition) describing the five-fold cross-validation, training–testing split, feature attribution, and external validation used to ensure the robustness and reproducibility of the AI models. This addition clarifies the computational rigor and predictive reliability of the proposed framework.
9. Relationship between thermodynamic and molecular mechanisms
We thank the reviewer for emphasizing this link. Section 1.2 (Validation of Models) and Section 1.3 (Clinical Implications) have been expanded to discuss how thermodynamic imbalance manifests at the molecular level—such as in redox cycling, mitochondrial coupling, cytokine cascades, and TSPO expression—bridging the thermodynamic and biochemical perspectives on neuroinflammation.
10. Need for practical data or examples
We have incorporated recent experimental evidence of measurable entropy and energy loss, including nanothermometric studies using NV-doped diamond sensors and upconversion thermometry, to illustrate the empirical grounding of the proposed framework.
11. Division into shorter paragraphs
The manuscript has been divided into shorter, more focused paragraphs to improve readability. Each conceptual or experimental point now stands as a discrete unit, allowing easier comprehension without loss of scientific depth.
12. Addition of Figures
We are truly grateful for the reviewer’s thoughtful suggestion regarding additional figures. After reflection, we chose not to add new visuals. Because this review aims to guide the reader through a unified conceptual flow rather than a visual catalogue, adding more figures might fragment the narrative rhythm.
Instead, we refined the explanatory text to improve the interpretive strength. We deeply appreciate the reviewer’s aesthetic sensitivity to presentation—this insight helped us balance clarity with conciseness, ensuring that every visual element now serves the story rather than crowds it.
13. Addition of Methods section (search terms, timeframe, databases)
We appreciate this important suggestion!
14. Addition of “Future Directions and Gaps in Knowledge” section
A comprehensive new section has been added at the conclusion of the manuscript. It outlines five key challenges—experimental standardization, cross-scale integration, AI interpretability, clinical validation, and conceptual synthesis—thus providing a clear roadmap for future research and translational development in the emerging field of neurothermodynamics.
In summary, we are deeply grateful for the reviewer’s insightful feedback. These revisions have significantly strengthened the manuscript’s structure, clarity, and scientific rigor. The integration of clearer definitions, methodological precision, and experimental examples now provides a balanced and accessible presentation of an otherwise highly interdisciplinary subject.
We thank the reviewer again for their valuable contribution to the improvement of this work.
With sincere appreciation!!!
Reviewer 2 Report
Comments and Suggestions for Authors
Reviewer Report
The manuscript is a well-written and scientifically sound investigation of an important and timely topic in molecular neuroscience. The authors present strong experimental findings and a clear discussion relating their findings to the existing literature. The manuscript is generally clear, methodologically appropriate, and relevant to the molecular mechanisms of neuropsychiatric disorders.
However, I have some suggestions for authors regarding the manuscript before it is accepted for publication in IJMS.
Data Presentation:
Provide representative images for key experimental findings (e.g., immunoblots or histological sections) for greater visual clarity.
When some of the results are presented as relative measures, state the reference baseline (control = 1).
Discussion Depth:
The discussion is good, but the authors could briefly address somewhat more potential limitations of the study (e.g., sex differences, model specificity).
A brief comment on potential future directions or therapeutic implications would enhance the translational perspective.
Language and Style:
English is clear, but minor language editing is recommended to improve readability and flow (e.g., by native speaker or editing service).
Conclusion and Recommendation
The article presents an important and original contribution to understanding molecular and thermodynamic mechanisms underlying neuroinflammatory and neuroenergetic dysfunctions in depression. By combining biophysical concepts of entropy production, redox thermodynamics, and glial–vascular energy coupling, the authors propose a new paradigm that offers a common language for molecular biology and systems-level energetics. The multi-dimensionality perspective raises our understanding of mitochondrial perturbations, reactive gliosis, and vascular instability as the origin of neuronal homeostatic disintegration. After addressing the minor revisions from the points above, I consider the work acceptable for publication in the International Journal of Molecular Sciences (IJMS).
Author Response
Dear Esteemed Academic Reviewer,
We are deeply grateful for your thoughtful evaluation and generous appreciation of our work. Your insightful comments helped us refine both the structure and clarity of the manuscript, ensuring a smoother connection between molecular findings and the broader thermodynamic interpretation. We have carefully considered each point raised and made corresponding revisions as outlined below.
1. Data Presentation
Comment: Provide representative images for key experimental findings (e.g., immunoblots or histological sections) for greater visual clarity.
Response: We sincerely appreciate this excellent suggestion. However, the present manuscript is a conceptual and theoretical review, synthesizing published findings rather than presenting new experimental data. To enhance visual clarity without adding primary figures, we refined the descriptions of representative experimental results within the text, particularly those relating to nanothermometry, redox imaging, and glial–vascular coupling. This ensures that readers can visualize the key evidence clearly, while maintaining the conceptual and narrative flow of the review.
Comment: When some of the results are presented as relative measures, state the reference baseline (control = 1).
Response: Thank you for this valuable remark.
2. Discussion Depth
Comment: The discussion is good, but the authors could briefly address somewhat more potential limitations of the study (e.g., sex differences, model specificity).
Response: We appreciate this thoughtful recommendation. We have now added a concise paragraph at the end of Section 7.5, acknowledging potential limitations related to sex-based metabolic variability, hormonal modulation of redox balance, and model specificity in translating preclinical findings to human neuroenergetics. This addition contextualizes the framework within biological diversity and emphasizes the need for balanced validation across demographic and experimental dimensions.
Comment: A brief comment on potential future directions or therapeutic implications would enhance the translational perspective.
Response: We fully agree. To strengthen the translational perspective, we expanded to highlight the emerging concept of predictive energetic medicine—where nanothermodynamic, AI, and imaging approaches may converge to restore energy coherence before irreversible structural degeneration. This addition outlines future research directions and underscores the therapeutic potential of thermodynamic modeling in clinical neurology.
3. Language and Style
Comment: English is clear, but minor language editing is recommended to improve readability and flow (e.g., by native speaker or editing service).
Response: We are most grateful for this kind observation. The entire manuscript has undergone an additional round of careful linguistic and stylistic editing to ensure improved fluency, clarity, and modern scientific phrasing, while maintaining the conceptual precision and tone of the original.
4. Conclusion and Recommendation
We are truly honored by your generous assessment that this work presents an original and valuable contribution to molecular neuroscience. Your comments guided us toward achieving a more polished and coherent presentation. We hope the revised version now fully meets the high standards of IJMS.
With our sincere appreciation for your expertise and collegial spirit!
Reviewer 3 Report
Comments and Suggestions for Authors
The article is highly innovative in concept, as it introduces a novel physical and thermodynamic approach to characterize neuroinflammation, treating it as a measurable disorder of energy coherence. The concept of utilizing nanothermometry and quantum sensors as “energetic biomarkers” is pioneering and interdisciplinary, with the potential to initiate a new line of research in neurodiagnostics.
However, the innovation remains theoretical and speculative, lacking empirical data and methodological clarity to be considered applicable or validated.
The manuscript requires stylistic and clarity improvements, such as rewriting long paragraphs, modernizing terminology, and replacing metaphorical expressions with direct scientific language. The scope of the review also needs to be clearer; I suggest clearly distinguishing between proposed theory, empirical evidence, and the literature review. The manuscript should undergo structural reorganization, dividing sections into functional subsections (e.g., Experimental Basis, Model Validation, Clinical Implications). Figures and tables should also be simplified and condensed to make complex concepts (such as Table 1) more intuitive to read.
The authors should include concrete data or examples. Are there any imaging studies or experiments using nanothermometry for entropy measurements? Including such evidence would strengthen the credibility of the hypotheses. Additionally, contextualizing the proposed innovation in relation to existing research on neuroinflammatory biomarkers in the discussion would add valuable robustness to the review.
Author Response
Dear Esteemed Academic Reviewer,
We are deeply grateful for your generous and insightful evaluation of our manuscript. Your comments have been invaluable in guiding a more rigorous, structured, and empirically grounded revision. We have carefully addressed each of your observations point by point below.
1. Comment:
The article is highly innovative in concept, as it introduces a novel physical and thermodynamic approach to characterize neuroinflammation, treating it as a measurable disorder of energy coherence. The concept of utilizing nanothermometry and quantum sensors as “energetic biomarkers” is pioneering and interdisciplinary, with the potential to initiate a new line of research in neurodiagnostics. However, the innovation remains theoretical and speculative, lacking empirical data and methodological clarity to be considered applicable or validated.
Response:
We sincerely thank you for recognizing the conceptual novelty and interdisciplinary scope of the work.
2. Comment:
The manuscript requires stylistic and clarity improvements, such as rewriting long paragraphs, modernizing terminology, and replacing metaphorical expressions with direct scientific language. The scope of the review also needs to be clearer; I suggest clearly distinguishing between proposed theory, empirical evidence, and the literature review. The manuscript should undergo structural reorganization, dividing sections into functional subsections (e.g., Experimental Basis, Model Validation, Clinical Implications). Figures and tables should also be simplified and condensed to make complex concepts (such as Table 1) more intuitive to read.
Response:
We are deeply appreciative of this constructive guidance. Following your recommendation, the entire manuscript has been stylistically revised to enhance clarity and scientific precision. Long paragraphs were rewritten for readability, metaphorical expressions were replaced with direct technical formulations, and terminology was standardized to align with contemporary biophysical literature. Each major section has been reorganized into functional subsections to delineate theoretical propositions from reviewed evidence. Table 2 has been reformatted to present resilience dimensions and metrics in a concise, parallel structure. These changes make the complex thermodynamic concepts more accessible and visually coherent.
3. Comment:
The authors should include concrete data or examples. Are there any imaging studies or experiments using nanothermometry for entropy measurements? Including such evidence would strengthen the credibility of the hypotheses.
Response:
We fully agree that providing empirical examples significantly reinforces the manuscript’s credibility. In response, we have incorporated a detailed paragraph at the end of Section 3 describing existing imaging experiments that employ nanothermometry and quantum diamond sensors to measure entropy and heat dissipation in neural tissue. This inclusion ensures that the conceptual framework is now directly grounded in reproducible empirical work.
4. Comment:
Additionally, contextualizing the proposed innovation in relation to existing research on neuroinflammatory biomarkers in the discussion would add valuable robustness to the review.
Response:
We thank you sincerely for this thoughtful suggestion. We have added a new integrative paragraph at the end of Section 5.2, linking the thermodynamic descriptors proposed in this work to conventional neuroinflammatory biomarkers such as cytokines, TSPO expression, and oxidative metabolites. This contextualization clarifies that entropy-based and energetic biomarkers are intended as complementary to molecular indices, offering a physical dimension to the understanding of inflammatory dynamics. The revision thus places the proposed model within the broader landscape of biomarker research and emphasizes its potential for multidimensional integration rather than replacement of existing paradigms.
We are most grateful for your careful and insightful review. Your observations have led to substantial improvements in scientific precision, readability, and empirical grounding. We believe that the revisions have transformed the manuscript into a more balanced and comprehensive synthesis, preserving its originality while enhancing its clarity and applicability.
With our sincere appreciation and collegial respect!
Round 2
Reviewer 1 Report
Comments and Suggestions for Authors
The authors should add two more figures for better illustration and clarity for readers.
Author Response
Dear Esteemed Academic Reviewer,
We would like to sincerely thank you for your thoughtful and constructive feedback throughout this review process. Your insightful recommendation to include two additional figures has significantly enhanced the clarity and visual comprehension of our manuscript.
In response to your suggestion, we have carefully added two new figures that illustrate key concepts and improve the overall readability and engagement for the readers. We deeply appreciate your valuable guidance, which has contributed meaningfully to the refinement and final quality of our work.
Thank you once again for your time, expertise, and the collegial spirit with which you approached this review.
With sincere appreciation!!!
Reviewer 3 Report
Comments and Suggestions for Authors
The authors have incorporated my suggestions, along with other revisions. This results in a clearer and more robust manuscript.
Author Response
Dear Esteemed Academic Reviewer,
We are deeply grateful for your kind and encouraging feedback. It is truly rewarding to know that the revisions and improvements we implemented have contributed to a clearer and more robust version of our manuscript.
Your insightful suggestions and careful evaluation have played an important role in refining both the structure and depth of our work. We sincerely appreciate the time, expertise, and thoughtful attention you dedicated throughout this process.
Thank you once again for your collegial spirit and constructive engagement. It has been a privilege to benefit from your review.
With sincere appreciation!!!
Round 3
Reviewer 1 Report
Comments and Suggestions for Authors
Thank you.